# Beneficial Physiological and Metabolic Effects with Acute Intake of New Zealand Blackcurrant Extract during 4 h of Indoor Cycling in a Male Ironman Athlete: A Case Study

**DOI:** 10.3390/jfmk9030141

**Published:** 2024-08-21

**Authors:** Mark E. T. Willems, Tilly J. Spurr, Jonathan Lacey, Andrew R. Briggs

**Affiliations:** 1Institute of Applied Sciences, University of Chichester, College Lane, Chichester PO19 6PE, UK; m.spurr@chi.ac.uk (T.J.S.); a.briggs@chi.ac.uk (A.R.B.); 2St Richard’s Hospital, Spitalfield Lane, Chichester PO19 6SE, UK

**Keywords:** anthocyanins, energy metabolism, supplementation, endurance exercise

## Abstract

New Zealand blackcurrant (NZBC) is known to alter exercise-induced physiological and metabolic responses with chronic (i.e., 7 days) dosing. We examined the effects of acute intake of New Zealand blackcurrant (NZBC) extract on 4 h indoor cycling-induced physiological and metabolic responses in a male amateur Ironman athlete (age: 49 years; BMI: 24.3 kg·m^−2^; $\dot{V}$O_2max_: 58.6 mL·kg^−1^·min^−1^; maximal aerobic power: 400 W; history: 14 Ironman events in 16 years) three weeks before competition. Indirect calorimetry was used and heart rate was recorded at 30 min intervals during 4 h indoor (~22.4 °C, relative humidity: ~55%) constant power (165 W) cycling on a Trek Bontrager connected to a Kickr smart trainer. Blood lactate and rating of perceived exertion (RPE) were taken at 60 min intervals. Study was a single-blind placebo-controlled study with capsules (4 × 105 mg anthocyanins) taken 2 h before starting the 4 h of cycling. Water was allowed ad libitum with personalised consumption of gels [a total of eight with three with caffeine (100 mg)], two bananas and 8 × electrolyte capsules (each 250 mg sodium and 125 mg potassium) at personalised time-points. With NZBC extract (CurraNZ), during 4 h of cycling (mean of 8 measurements), minute ventilation was 8% lower than placebo. In addition, there was no difference for oxygen uptake, with carbon dioxide production found to be 4% lower with NZBC extract. With the NZBC extract, the ventilatory equivalents were lower for oxygen and carbon dioxide by 5.5% and 3.7%; heart rate was lower by 10 beats·min^−1^; lactate was 40% different with lower lactate at 2, 3 and 4 h; RPE was lower at 2, 3 and 4 h; and carbohydrate oxidation was 11% lower. With NZBC extract, there was a trend for fat oxidation to be higher by 13% (*p* = 0.096), with the respiratory exchange ratio being lower by 0.02 units. Acute intake of New Zealand blackcurrant extract (420 mg anthocyanins) provided beneficial physiological and metabolic responses during 4 h of indoor constant power cycling in a male amateur Ironman athlete 3 weeks before a competition. Future work is required to address whether acute and chronic dosing strategies with New Zealand blackcurrant provide a nutritional ergogenic effect for Ironman athletes to enhance swimming, cycling and running performance.

## 1. Introduction

The Ironman triathlon is a single-day exercise endurance event with subsequent performance of 3.9 km swimming, 180.2 km cycling and 42.2 km running. Preparation for an Ironman triathlon requires a high physical training volume over months [1], with an average of 14.7 h per week reported for amateur athletes (n = 83, i.e., swimming: ~2.14 h·wk^−1^; cycling: ~6.75 h·wk^−1^; running: ~4.0 h·wk^−1^) [2]. In addition, preparation can also involve the adoption of a low-carbohydrate diet [3], carbohydrate supplementation strategies [4] and periodised training with low carbohydrate availability [5]. The physical training with an emphasis on regular endurance exercise will provide an enhanced enzymatic and non-enzymatic antioxidant defence [6]. In addition, the adaptations in the cardiovascular system by the prolonged endurance training regimes and the dietary and carbohydrate intake strategies may also enhance the capacity for fat oxidation by causing an increase in mitochondrial volume density [7] and increased capillarisation [8]. Interestingly, in male Ironman athletes, there was a significant negative correlation between race time and maximal fat oxidation (r^2^ = 0.12) [9]. However, it is well known that other physiological factors such as maximum oxygen uptake [9] and exercise economy [10] are more essential for race time and competitive completion during an Ironman event. During Ironman triathlon competition, carbohydrate intake is essential. It is also common for Ironman athletes to have adopted an intake strategy with caffeine, with the aim being to enhance endurance performance [11].

In Ironman athletes, laboratory testing has been focused on metabolic (i.e., maximal fat oxidation) [9] and physiological parameters (e.g., heart rate at ventilatory thresholds [12] to predict overall Ironman competitive performance. As far as we know, observations from laboratory testing on the effectiveness of ergogenic nutritional supplementation in Ironman athletes or other ultra-endurance athletes that are close to competition are absent. However, such testing will reveal whether highly trained individuals are responsive to the supplement but has no predictive value for Ironman competition. Recently, we examined the effects of 7-day intake of New Zealand blackcurrant (NZBC) extract in a male ultra-endurance athlete during 2 h of treadmill running [13]. The ultra-endurance male athlete was tested in the 6 weeks between competing in two 100-mile runs. In the male ultra-endurance athlete, 7-day intake of NZBC extract enhanced running-induced fat oxidation [13]. The case study by Willems and Briggs [13] was the first observation that an ultra-endurance athlete who trained to compete in 100-mile events could still respond with enhanced exercise-induced fat oxidation similar to what has been reported in recreationally active males with 2-week intake of NZBC extract (600 mg containing 210 mg of anthocyanins) during moderate-intensity walking [14]. New Zealand blackcurrant is an anthocyanin-rich fruit with a specific anthocyanin composition consisting of primarily delphinidins and cyanidins [15]. Anthocyanins and anthocyanin-induced metabolites are thought to have antioxidant and anti-inflammatory effects, and evidence is mounting for their health-promoting properties (e.g., [16,17]). However, research on the effects of anthocyanin intake during exercise is still limited [18]. In addition, the effectiveness of the intake of NZBC has been examined in dosing studies that lasted mainly from 7 to 14 days (e.g., [14,19]) and up to 5 weeks [20]. Studies on the acute effects NZBC extract on metabolic and physiological responses during exercise are limited. Studies have reported that the acute intake of a blackcurrant extract affected cardiovascular responses during prolonged sitting with 1.84 [21] and 1.87 mg total anthocyanins per kg bodyweight^−1^ [22].

In the present study, we examined the effects of the acute intake of NZBC extract on metabolic and physiological responses during 4 h of constant power cycling in a case study on an Ironman athlete three weeks before competing in the Ironman Callela-Barcelona (1 October 2023). For this study, we examined the effect of an acute intake of NZBC extract as observations in multiple-day dosing studies may still be associated with the intake of the final dose (e.g., [13]). Compared with the dosing strategy of other performance-enhancing supplements (e.g., creatine and caffeine), the optimal dosing strategy for ergogenic effects by intake of anthocyanins is still unknown. However, in a home-based study on the acute effects of 900 mg of anthocyanin-rich NZBC extract in a cohort of trained cyclists, it was observed that the faster cyclists did not enhance their 16.1 km cycling time-trial performances [23]. It was suggested that higher acute intake may be required. In addition, Moss et al. [24] reported no acute effects with 900 mg of NZBC extract on the metabolic and physiological responses during 5 km treadmill running. An acute dosing effect with a higher intake than 900 mg of blackcurrant anthocyanins has not been addressed in case or cohort studies in ultra-endurance athletes. A study in a cohort of Ironman athletes, three weeks before a competition, would be logistically challenging considering that some of the requirements for a sports nutrition study such as same-time-of-day testing, having a placebo-controlled design and considering ongoing physical training and tapering programmes in a cohort would be difficult to meet.

Therefore, in the present case study, we examined the effects on exercise-induced physiological and metabolic responses in a male Ironman athlete during scheduled training sessions of 4 h of cycling by the acute intake of 1200 mg of NZBC extract containing 420 mg of anthocyanins.

## 2. Materials and Methods

### 2.1. Participant and Ethical Approval

A male amateur Ironman athlete (history: 14 Ironman events in 16 years) volunteered. Convenience sampling was used; see Table 1 for participant’s characteristics. Written informed consent was obtained after agreement with the participant on the experimental procedures, potential risks and right to withdrawal. The participant was scheduled to perform 4 h cycling rides as part of the preparation training for the Ironman event. The participant completed a health history questionnaire and had no supplement or medicine use that could interfere with the metabolic and physiological outcome measures of the study. This study was approved according to the policy guidelines of the Research Ethics Committee of the University of Chichester (United Kingdom) (ethical approval code: 2223_67, approval date: 25 August 2023), with the procedures and protocol adhering to the 2013 Declaration of Helsinki.

### 2.2. Experimental Design and Cycling Tests

The participant attended the Exercise Physiology laboratory at the University of Chichester (United Kingdom) for three visits. In the first visit, a body composition measurement was taken for body fat percentage (InBody 770, InBody UK, Coalville, UK). The participant completed an incremental cycling protocol (8 stages of 4 min with 2 min rest periods, starting power at 50 W with 30 W increments (adapted from González-Haro et al. [25]) and a self-selected constant pedaling rate between 70 and 90 rpm). Expired air samples were collected for each stage. Blood samples were taken upon the completion of each stage to determine the cycling power at a blood lactate concentration of 4 mmol·L^−1^ [26]. After a 20 min rest, the participant then completed an incremental cycling protocol (starting power 50 W with increments of 30 W every minute) (adapted from Bailey et al. [27]) with expired air collection in Douglas bags in the final minutes of the test. Expired air was analysed with indirect calorimetry techniques to assess maximal oxygen uptake. All cycling testing was on a participant-owned Trek Speed Concept Bontrager (and used in the Ironman Callela-Barcelona 2023) connected to a Kickr Smart Trainer (Wahoo Kickr Smart Trainer, Wahoo Fitness, London, UK). For visits two and three, this study used a placebo-controlled, randomised (using a coin) cross-over single-blind design. As there were no time-performance measurements, there was no tester bias in the testing environment. Visit two was the placebo condition with testing on the 8th September 2023 (23 days before competition), and visit three was the NZBC extract on the 15th September 2023 (16 days before competition). With participant agreement and consideration of power observations during preparatory cycle training, it was decided to perform 4 h of cycling at a constant power of 165 W (cycle application Zwift^®^, Zwift Inc., Long Beach, CA, USA) with self-selected constant pedalling frequency. For visits two and three, the participant arrived in the laboratory at 10 a.m., allowing for a habitual breakfast 3 h before and compulsory intake of the NZBC extract at 9 a.m. The participant did not exercise on the day before visits two and three. Dietary intake was recorded for 48 h prior to visits two and three, and the related data are presented in Table 2. A food frequency questionnaire with foods and drinks listed in the Phenol-Explorer database [28] was completed to estimate habitual anthocyanin intake.

### 2.3. Supplementation Strategy during 4 h of Indoor Cycling

During the 4 h constant power cycling test, the participant used a personalised dosing strategy (i.e., 8 × 250 mg sodium and 8 × 125 mg potassium, 5 × 30 g of carbohydrate gels, 3 × 30 g of carbohydrate gels with 100 mg caffeine, and 4 × half banana and water ad libitum; see Figure 1 for the timepoints of intake during the 4 h cycling test).

### 2.4. Measurements during Visits for 4 h of Indoor Cycling

Figure 2 shows the measurements during the 4 h of cycling. During the 4 h of cycling with constant power, expired air was collected with 2 min Douglas bag collections every 30 min (8 collections). Heart rate was measured (RS400, Polar Electro UK Ltd., Warwick, UK) and averaged during the 2 min expired air collections. Blood samples were taken every 60 min (4 samples) and measured in duplicate for lactate (Biosen C-Line, EKF Diagnostics, Cardiff, UK). A rating of perceived exertion (Borg: 6–20 scale) was taken every 60 min. For the measurements of respiratory parameters, expired air was analysed for fractions of oxygen and carbon dioxide using a calibrated oxygen and carbon dioxide gas analyser (Servomex 1440, Servomex plc, Crowborough, UK). During the 2 min expired air collections, inspired fractions of oxygen and carbon dioxide were also measured. The volume of expired air was measured with a calibrated dry gas meter (Harvard Apparatus Ltd., Edenbridge, UK) with simultaneous measurement of expired air temperature during Douglas bag evacuation for gas volume temperature corrections. Barometric pressure was measured using a mercury barometer at the start of each testing session. Environmental conditions for ambient temperature and humidity (Kestrel Meter 5400 Heat Stress Tracker, Kestrel Meters, Boothwyn, PA, USA) were measured during expired air collections (8 measurements).

### 2.5. Placebo and New Zealand Blackcurrant Extract Intake

Two hours before starting the 4 h cycling test, the participant athlete was dosed with 4 capsules of NZBC extract or an identical-looking placebo (4 × 300 mg microcrystalline cellulose M102). Each blackcurrant capsule contains a 300 mg NZBC extract (CurraNZ™, Health Currancy Ltd., Surrey, UK) containing 105 mg of anthocyanins. According to company information, the anthocyanin composition of each 300 mg capsule was 35–50% delphinidin-3-rutinoside, 5–20% delphinidin-3-glucoside, 30–45% cyanidin-3-rutinoside and 3–10% cyanidin-3-glucoside. Anthocyanins in the blood plasma reached peak levels a few hours after intake [21].

### 2.6. Data Calculations and Statistical Analysis

Exercise-induced whole-body fat and carbohydrate oxidation during the 4 h cycling test were calculated with equations from Frayn et al. [29]. A paired two-tailed *t*-test was used for analysis of the 30 min measurements of minute ventilation, oxygen uptake, carbon dioxide production, ventilatory equivalents for oxygen and carbon dioxide, heart rate and pedaling frequency (i.e., 8 time points for each parameter) (GraphPad Prism v5 for Windows). Data were reported as mean ± SD and 95% confidence intervals, calculated from the 8 time-point measurements during the 4 h cycling test. Significance was accepted at *p* < 0.05.

## 3. Results

### 3.1. Dietary Intake

In Table 2, we present the daily dietary and energy intakes in the 48 h before testing for 4 h of cycling in the placebo and NZBC extract conditions.

The higher energy intake in the placebo condition was mainly due to consumption of non-alcoholic ginger beer. The Ironman athlete was self-informed on dietary intake. Dietary records of the habitual breakfast 3 h before testing indicated no potentially meaningful intake of caffeine. The habitual dietary anthocyanin intake (i.e., without the intake of NZBC extract) for the male Ironman athlete was estimated to be 13.4 mg·day^−1^.

### 3.2. Body Mass and Ad Libitum Water Intake

During the 4 h constant power cycling test at 165 W, the participant covered ~75% of the competitive Ironman cycling distance (placebo: speed: 9.297 m·s^−1^, cadence: 83.6 rpm, distance: 133.88 km; NZBC extract: speed: 9.300 m·s^−1^, cadence: 80.6 rpm, distance: 133.94 km). There were no differences for temperature and humidity in the laboratory between the conditions that could potentially account for a difference in the exercise responses (placebo: 22.8 ± 0.9 °C, 53 ± 1%; NZBC extract: 22.2 ± 0.2 °C, 58 ± 5%). Body mass change was −2.1 kg in the placebo condition (83.2 to 81.1 kg) and −1.4 kg in the NZBC extract condition (82.5 to 81.1 kg), with ad libitum water intake for the placebo and NZBC extract conditions of 2135 and 2983 mL.

### 3.3. Physiological Responses during the 4 h of Cycling

Table 3 provides the physiological responses during 4 h of cycling in the placebo and NZBC extract condition. In the NZBC extract condition, the heart rate was 10 beats·min^−1^ lower during the 4 h of cycling (placebo: 95%CI [130, 149 beats·min^−1^]; NZBC extract: 95%CI [124, 135 beats·min^−1^]). Minute ventilation was 7.7% lower (placebo: 95%CI [56.6, 63.4 L·min^−1^], NZBC extract: 95%CI [53.9, 56.5 L·min^−1^]). Oxygen uptake was not affected (placebo: 95%CI [2.49, 2.60 L·min^−1^], NZBC extract: 95%CI [2.41, 2.55 L·min^−1^]). Carbon dioxide production was 4.2% lower (placebo: 95%CI [2.18, 2.30 L·min^−1^], NZBC extract: 95%CI [2.08, 2.21 L·min^−1^]). The ventilatory equivalent for oxygen was 5.5% lower (placebo: 95%CI [27.1, 29.7], NZBC extract: 95%CI [26.4, 27.1]). The ventilatory equivalent for carbon dioxide was 3.7% lower (placebo: 95%CI [31.0, 33.3], NZBC extract: 95%CI [30.7, 31.2]).

### 3.4. Metabolic Responses and Rating of Perceived Exertion during 4 h of Cycling

NZBC extract lowered the respiratory exchange ratio by 0.02 units during the 4 h of cycling (placebo: 0.88 ± 0.01, 95%CI [0.87, 0.89]; NZBC extract: 0.86 ± 0.01, 95%CI [0.85, 0.87], *p* = 0.042). In addition, carbohydrate oxidation was lowered by 11% (placebo: 2.04 ± 0.17 g·min^−1^, 95%CI [1.90, 2.18 g·min^−1^]; NZBC extract: 1.80 ± 0.13 g·min^−1^, 95%CI [1.69, 1.91 g·min^−1^]; *p* = 0.025) (Figure 3a). For fat oxidation, there may have been an increase of 13% (placebo: 0.50 ± 0.06 g·min^−1^, 95%CI [0.45, 0.55 g·min^−1^]; NZBC extract: 0.56 ± 0.05 g·min^−1^, 95%CI [0.52, 0.61 g·min^−1^]; *p* = 0.096) (Figure 3b). For lactate, Figure 3c shows cycling-induced lactate responses with overall 40% lower values in the NZBC extract condition. The acute intake of NZBC extract with 420 mg of anthocyanins was able to affect the mechanisms that regulate substrate oxidation during 4 h of cycling with personalised intake of sodium, carbohydrate and caffeine in a male Ironman athlete. Figure 3d shows the rating of perceived exertion RPE during 4 h of cycling with lower values at 2, 3 and 4 h.

## 4. Discussion

The present study proved, for a male (age: 49 years) amateur Ironman athlete, that, in comparison to placebo, an acute intake of 420 mg of anthocyanins from New Zealand blackcurrant extract 2 h before undertaking 4 h of constant power cycling altered physiological (e.g., lower heart rate) and metabolic responses (e.g., enhanced fat oxidation). Physical training to allow competition in an Ironman event will normally require >12 h per week of swimming, cycling and running over a period of months (for a review, see Knechtle et al. [30]). The present case study was able to test an amateur male Ironman athlete (age: 49 years) for the effectiveness of an acute intake of 420 mg of anthocyanins from a New Zealand blackcurrant extract three weeks before taking part in an Ironman competition in Barcelona, Spain (2023, finish time: 15:32:32 (swim: 1:26:48; cycle: 6:23:27; run: 7:24:32). It was of interest, therefore, to examine whether New Zealand blackcurrant can affect the highly adapted cardiorespiratory and neuromuscular functions that allow Ironman completion. The male amateur Ironman athlete in the present study had a $\dot{V}$O_2max_ of 58.6 mL·kg^−1^·min^−1^, and this was comparable to the reported cohort $\dot{V}$O_2peak_ data of younger [age: 35 ± 1 yr, 58.7 ± 0.7 mL·kg^−1^·min^−1^ [range 43.9–72.5]] male Ironman athletes [9].

In general, the effectiveness of nutritional ergogenic aids should be examined in different cohorts and environmental conditions as many factors, e.g., genetics, ethnicity, habitual dietary intake, training status, environmental temperature and altitude may affect the physiological and metabolic responses (e.g., [31,32]). In previous studies on the effects of New Zealand blackcurrant powder and extract, 7-day dosing was mostly used with 105, 210 or 315 mg of blackcurrant anthocyanins to examine physiological, metabolic and performance responses in cohorts of recreationally active and endurance-trained individuals (e.g., [14,19,33,34]). These cohort studies have provided observations on the effectiveness of New Zealand blackcurrant to enhance exercise-induced fat oxidation. In a case study of a male ultra-endurance runner, enhanced fat oxidation during a 2 h treadmill run (speed: 10.5 km·h^−1^) was also shown with 7-day dosing with 210 mg anthocyanins in the 6 weeks between two 100-mile running events [13]. In a cohort study by Montanari et al. [23], an acute intake of 315 mg was used in a home-based study on 16.1 km cycling time-trial performance, and no metabolic responses could be measured. In Montanari et al. [23], the slower cyclists showed enhanced 16.1 km cycling time-trial performance. However, a recent study by Moss et al. [24] provided no acute effects of 315 mg of blackcurrant anthocyanins on physiological and metabolic responses at the lactate threshold but with a 5 km performance effect in trained male runners. The dosing strategy for the acute intake of blackcurrant anthocyanins recommended to provide beneficial physiological and metabolic efforts for different cohorts is not known. In the present study, there was not a clear effect on 4 h cycling test-induced fat oxidation with the acute intake of 420 mg of blackcurrant anthocyanins. Although the participant consumed caffeine at 150, 180 and 210 min during the 4 h of cycling, the amount each time was 1.2 mg per kg body weight. This dose of caffeine is not known to affect exercise-induced physiological and metabolic responses. It is possible that the total intake of 240 g of carbohydrate at regular intervals somewhat blunted the potential fat oxidation response. However, we observed a decreased contribution of carbohydrate oxidation to the energy demands, thus suggesting that the metabolic responses were affected. In addition, 2 h into the 4 h of cycling, we observed that lactate became lower, providing additional evidence that metabolic responses were affected. Although whole-body oxygen consumption did not seem to be affected, it is possible that local oxygen delivery to skeletal muscle fibres may have been enhanced due to vasodilatory effects by the blackcurrant intake [21], therefore depending less on metabolic pathways that contribute to the elevation of lactate during exercise [35]. In the present study, mean lactate in the placebo condition was 2.5 ± 0.6 mmol·L^−1^ and comparable to the study by Laursen et al. [12] with a cohort of male Ironman athletes (age: 35.8 ± 1.6 y, $\dot{V}$O_2peak_: 67.5 ± 1.0 mL·kg^−1^·min^−1^) with observed lactate values of 2.8 ± 0.4 mmol·L^−1^ during 5 h of cycling with an average power of 188 ± 9 W. Lower lactate values with the acute intake of New Zealand blackcurrant extract may suggest an ability to postpone fatigue mechanisms. In addition, it is possible that enhanced fat oxidation may result in some glycogen sparing [36]. We cannot exclude, however, the possibility that the observed metabolic responses were affected by ongoing training activities, but training sessions with overload progression were finished. Future work may address the performance-enhancing effects during long-distance cycling time-trials (e.g., 100-km [37]) with dosing strategies combining carbohydrates and blackcurrant anthocyanins.

Cohort studies have never reported on lower heart rate, minute ventilation and ventilatory equivalents with intake of New Zealand blackcurrant extract. For Ironman triathlon competitions, heart rate and ventilatory responses may be associated with the perceived exertion during the competition. Future work should recruit a cohort of male and female Ironman athletes to allow for the generalisation of the findings of the present study. In the present study, however, the acute intake of New Zealand blackcurrant extract had a beneficial effect on the ventilatory equivalents of oxygen and carbon dioxide together with lower perceived exertion. Lower ventilatory equivalents may be indicative of enhanced functional breathing [38] and, therefore, allowing a lower heart rate. In addition, it may be speculated that enhanced functional breathing can help competing ultra-endurance athletes to potentially avoid the development of fatigue in inspiratory muscles [39].

## 5. Conclusions

The acute intake of New Zealand blackcurrant extract (420 mg anthocyanins) provided beneficial physiological (lower heart rate, minute ventilation and ventilatory equivalents) and metabolic responses (high fat oxidation and lower lactate and carbohydrate oxidation) during 4 h of indoor constant power cycling in a male amateur Ironman athlete 3 weeks before a competition.

## Figures and Tables

**Figure 1 jfmk-09-00141-f001:**
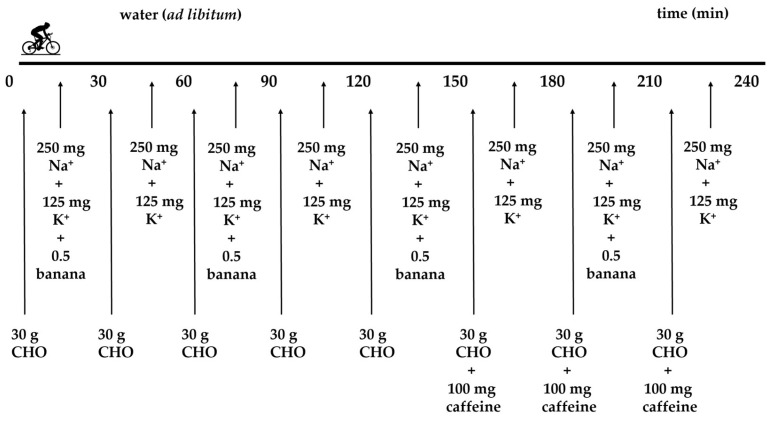
Dosing strategy of the male Ironman athlete during 4 h of constant power indoor cycling. Na^+^, sodium; K^+^, potassium; CHO, carbohydrate.

**Figure 2 jfmk-09-00141-f002:**
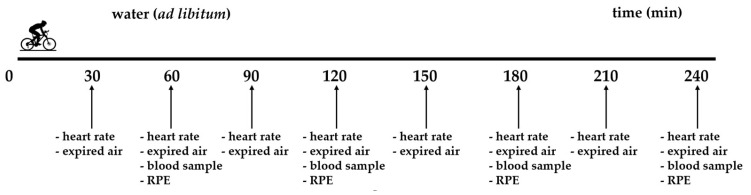
Measurements during the 4 h of constant power indoor cycling. RPE, rating of perceived exertion.

**Figure 3 jfmk-09-00141-f003:**
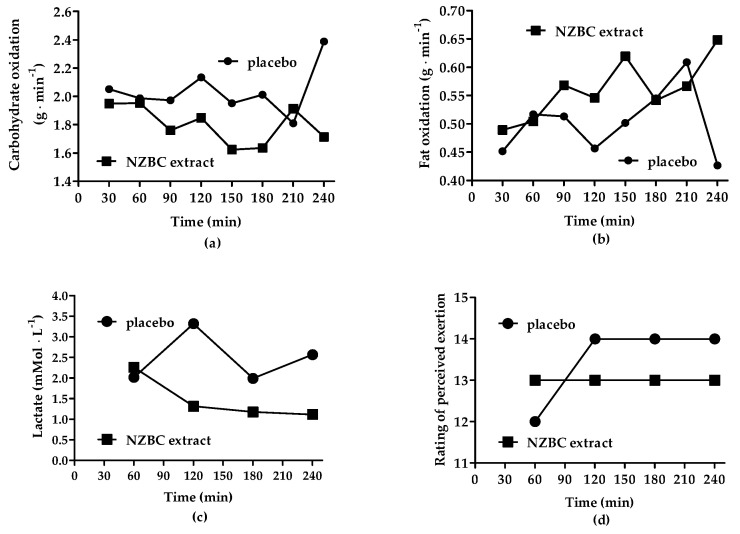
Carbohydrate oxidation (**a**), fat oxidation (**b**), blood lactate (**c**) and the rating of perceived exertion (**d**) during the 4 h of cycling. NZBC, New Zealand blackcurrant.

**Table 1 jfmk-09-00141-t001:** The characteristics of the male Ironman athlete.

Age (years)	49
Body mass (kg)	83.3
Body mass index (kg·m^−2^)	24.3
Body fat (%)	13.5
Maximum oxygen uptake (mL·kg^−1^·min^−1^)	58.6
Maximal aerobic power (Watts)	400
Onset of blood lactate accumulation (4 mmol·L^−1^) (Watts)	289

**Table 2 jfmk-09-00141-t002:** Daily dietary and energy intake in the 48 h before each experimental visit for the male Ironman athlete.

	Conditions	
	Placebo	NZBC Extract
Parameters		
Carbohydrate (g, %)	349, 49	270, 46
(g∙kg body mass^−1^)	4.2	3.2
Fat (g, %)	139, 19	106, 18
(g∙kg body mass^−1^)	1.7	1.3
Protein (g, %)	103, 32	93, 36
(g∙kg body mass^−1^)	1.2	1.1
Energy intake (kcal)	2876	2337

NZBC, New Zealand blackcurrant.

**Table 3 jfmk-09-00141-t003:** Physiological responses during 4 h of cycling under the placebo and NZBC extract conditions.

Parameter	Placebo	NZBC Extract	
Heart rate (beats·min^−1^)	140 ± 11	130 ± 6 *	*p* < 0.01
Minute ventilation (L·min^−1^)	60.0 ± 4.1	55.2 ± 1.6 *	*p* = 0.012
Oxygen uptake (L·min^−1^)	2.54 ± 0.07	2.49 ± 0.08	*p* = 0.11
Carbon dioxide production (L·min^−1^)	2.24 ± 0.07	2.15 ± 0.07 *	*p* = 0.027
Ventilatory equivalent (oxygen)	28.4 ± 1.5	26.8 ± 0.5 *	*p* = 0.023
Ventilatory equivalent (carbon dioxide)	32.2 ± 1.4	30.9 ± 0.4 *	*p* = 0.033

NZBC, New Zealand blackcurrant. *, a difference between conditions.

## Data Availability

Data are available on reasonable request.

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
