# Peer review of "Beneficial Physiological and Metabolic Effects with Acute Intake of New Zealand Blackcurrant Extract during 4 h of Indoor Cycling in a Male Ironman Athlete: A Case Study"

_jfmk, 2024, doi:10.3390/jfmk9030141_

Round 1
Reviewer 1 Report
Comments and Suggestions for Authors
Firstly, I would like to thank you for reviewing this manuscript titled Beneficial Physiological and Metabolic Effects with Acute Ingestion of New Zealand Blackcurrant Extract During 4h Indoor Cycling 3 in a Male Ironman Athlete: A Case Study; and congratulate the authors for their work.
Although the authors produced a detailed summary; It is recommended to include at least one background sentence in the summary.
In short, it is recommended to remove evil characteristics from the target. This aspect must be described in the method.
In the summary it is not clear how the subject is compared, other than with the same subject in different measurements or with the other subject. If there are comments like there is less carbon dioxide production, we don't know what it compares to.
The summary does not clearly indicate the implementation process of New Zealand blackcurrant extract; No mention of gels, bananas, etc.
It is recommended not to use the same keywords that appear in the title.
Line 83, there are two points. Review the complete document.
On the other hand, the amount of NZBC administration must be primarily justified.
The introduction is considered adequate and well-founded. However, behind the objective it is stated that “an acute dosage effect with 420 mg blackcurrant anthocyanins-95 years2 has not been addressed in case or cohort studies2, a statement that must be explained in advance.
Hensinki Delcaradción must be incorporated.
The authors describe the experimental design and the cycling test, however, greater detail and justification of the chosen tests must be incorporated. Therefore, a greater number of citations should be included.
The information and quotes included in point 2.4 should be expanded. Justify the dose as much as possible and explain if it occurs at two different times, if it is compared in two different sessions. Of course, it doesn't fall.
Apply the comments noted above to the discussion.
Do not include results in the discussion and focus on commenting on them and arguing with previous findings.
Reviewer 2 Report
Comments and Suggestions for Authors
In this study, authors analyzed the physiological and metabolic effects of New Zealand Blackcurrant extract on 4-hours indoor cycling for an Iron Man.
The purpose of this study can be seen to analyze the effects of the New Zealand Blackcurrant extract. For this purpose, a male Iron Man was selected and measures such as calorie, heart rate, and respiratory rate were introduced to derive beneficial effects of the extract.
However, in the analysis process of this paper, a very special case was presented without introducing a general and objective analysis process. In other words, it is very difficult to generalize the results of this study by one participant, 3 weeks of observation, 4 hours of indoor cycling, etc.
Therefore, the authors of this study need to reinforce the study by introducing an objective environment such as more diverse experiment participants and longer observation periods.
Reviewer 3 Report
Comments and Suggestions for Authors
Dear Editor:
This is an interesting work where the authors have carried out an important study on the physiological and metabolic effects of ingesting currant extract on athletes.
However, in order to be considered for publication, authors must take into account the following considerations:
Introduction: the units must be described in the International System (lines 37 and 40) remove miles. At the same time, the units of the international system k.w/k are not correct, please modify.
Line 83, withdraw (01/10/2023)..
Lines 93-94, include a table with these nutritional values and contributions to make it more intuitive and easy to view.
Materials and methods: line 98, the authors must justify the experimental design and why that type of sample was selected. (age 42 years).
Figures 1 and 2 should improve resolution and use the same font in the tables as used in the document.
In Table 1 the units are doubtful. The authors refer to mass/mass percentage (m/m). How many readings were done? the mean and standard deviation must be included.
Reviewer 4 Report
Comments and Suggestions for Authors
This study examines the effects of acute ingestion of New Zealand blackcurrant (NZBC) on physiological measures of physical exercise and glucose metabolism in male amateur Ironman athletes. The design of the study is a case study with one subject. The study was conducted in a single-blind, placebo-controlled format. Although this is a single case study paper, I believe that the format and discussion of the study as a case study have been well considered and that the study is valuable enough to serve as a basis for future research.
In addition, this study suggests the possibility of acute ingestion. I believe that this is a fascinating study that will allow for the inclusion of a wide range of subjects in future studies, from healthy subjects and the elderly to citizen athletes and top athletes.
Major Comments
This study is a single case study. In a case study, it is essential to identify the demographics of the subjects. Subject attributes are clearly stated in the text (L98-109). One option to make the subject attributes more straightforward is to present them in a table format. For example, I believe it would be possible to indicate subject attributes in accordance with Table 1. I appreciate your consideration.
This study is a case study and has implications as a pilot study for future quantitative studies. I can also consider the transition to an intervention study. The discussion session (L309-312) discusses a few directions for future research.
I believe that more detailed descriptions of possible future research should be provided to clarify the purpose of this study and to clarify the direction of future research in the same line of research. Please consider.
Round 2
Reviewer 2 Report
Comments and Suggestions for Authors
In this study, it is difficult to generalize the research results by studying one person, but very meaningful results were derived through a very detailed analysis method.
In the future, it is strongly recommended that the number of participants must be increased and applied.
In addition, describe the motivation of the study more clearly and in detail in the introduction part(chapter 1).
Reviewer 3 Report
Comments and Suggestions for Authors
Dear editor, thank you very much. The authors have made the suggested changes and the article can now be considered for publication
Author Response
Thanks again for your time and anticipation.